# Heavy Metals and as in Ground Water, Surface Water, and Sediments of Dexing Giant Cu-Polymetallic Ore Cluster, East China

**Hanjiang Pan** [1], **Guohua Zhou** [1], **Rong Yang** [1,*], **Zhizhong Cheng** [2] **and Binbin Sun** [1]

1 Institute of Geophysical and Geochemical Exploration, China Geological Survey, Langfang 065000, China; phanjiang@mail.cgs.gov.cn (H.P.); zguohua@mail.cgs.gov.cn (G.Z.); sbinbin@mail.cgs.gov.cn (B.S.)
2 Development and Research Center, China Geological Survey, Beijing 100037, China; chengzhizhong69@163.com
* Correspondence: yangrong0516@163.com

**Abstract:** Heavy metals and As (HMs) pollution in mining areas are a widespread environmental concern. In this study, ground water, surface water, and sediment samples around the Dexing area, one of the largest Cu-polymetallic ore clusters in China, were collected to examine the concentrations and distributions of As, Cd, Cr, Cu, Hg, Pb, and Zn. Pollution indices, geo-accumulation index, and potential ecological risk index were used to estimate the pollution characteristics and ecological risk of HMs. The results show that the major pollutants in the surface water were Cd, Cu, Zn, and Pb, while the dominant ecological risk of HMs in the sediments originated from Cu, As, Hg, and Cd. Moreover, HMs in the surface water and sediments exhibited substantial spatial heterogeneity in the study area, indicating a severely disturbed environment due to mining activities. The proportions of HM pollutions were higher in the Dexing River and its tributaries than in the Le'an River and its tributaries. The surface water pollution was predominant at the tributaries closest to the mine area, while the sediment contamination has been expanded several kilometers downstream of the major rivers. Overall, the ecological risk of HMs was higher in the sediments than in the surface water.

**Keywords:** heavy metals; water; sediments; Dexing ore cluster; environmental risk

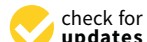



## 1. Introduction

Heavy metals and As (HMs) comprise the group of elements with a density larger than $4\ \mathrm{g\ cm^{-3}}$, including metals and metalloids [1]. HMs are reportedly carcinogenic and toxic to human and animals; they cause cancer, cardiovascular disease, neurotoxicity, and diabetes [2–6]. Mining areas, particularly non-ferrous metal mines, are characterized by severe HMs pollution. This can be attributed to the high background concentrations and extremely high concentrations of HMs. These concentrations are sometimes detected in surface water, sediments, and surface soils around unexploited deposits [7,8]. Moreover, mining activities can trigger the leaching of large quantities of HMs into the environment [9], particularly during opencast mining and smelting activities as they exert substantial environmental impact on soils and water streams [10]. The European industrial sector reported that mining and metal processing industries contribute to 48% of the total release of HMs contaminants [11]. Moreover, vast amounts of mining waste, such as tailings and waste rock, can cause long-term and large-scale HMs pollution, even decades after mines have been closed [12–14].

The generation and release of acid mine drainage (AMD) from the oxidation of sulfide ores/minerals is likely the major cause of HMs pollution around non-ferrous metal mines [15]. Most of AMD, which contain high concentrations of HMs, flows into river and groundwater system. Although the concentration can be low, HMs in water are primarily found in the water-soluble state, which can be easily absorbed and accumulated in organisms through food chain circulation, thereby threatening human health [16]. For example, untreated ground water is typically used as drinking water in rural China, and

large groups of populations are thus exposed to these threats. As the largest pool of HMs, sediments simultaneously act as a carrier and a secondary source of pollutants in the aquatic system [17–19]. Moreover, HMs concentrations in the sediment are typically four or five times higher than the overlying water [20]. When the environmental factors including Eh, pH, and organic matter change, the HMs in sediment may be released into the overlying water [21]. Therefore, sediments, surface water, and ground water are essential sources for evaluating the HMs contamination in the aquatic systems of mining areas.

The HMs contamination in some regions of China can be peculiarly acute. The Dexing area in Jiangxi Province is a giant polymetallic ore belt in China with numerous large porphyry copper, Pb–Zn polymetallic, and Au deposits [22]. The HMs pollution in the Dexing area has been a concern for a long time. For example, Yi [23] studied the water quality of main streams in Dexing area from 2002 to 2017 and showed that most HMs were high in the main streams of mining areas, but that the water quality of 2017 was better than before. Xiao et al. [24] found that heavy metal concentrations in river waters (dissolved), suspended solids and sediments all showed highly localized distribution patterns closely associated with two AMD-contaminated tributaries (Dawu and Ji), and are significantly different from the previous findings. Yin et al. [25] evaluated the pollution levels of HMs in the 4# tailing pond of Dexing copper mine and pointed out that Cd and Cu pollutions reached the levels of moderate and heavy, respectively. Teng et al. [26] determined the HMs concentrations in the surface water, sediments, soils and plants around the Dexing mining area, and the result indicated a highly localized distribution pattern closely associated with the two pollution sources along the Le'an River bank: one is strong acidity and a large amount of Cu in the drainage from the Dexing Cu mining area; and the other is the high concentration of Pb and Zn in the effluents released from many smelters, mining, processing and extracting activities in the riparian zone. For the crops, Zhou et al. [14] confirmed that the concentration of Cd in 27.8% paddy rice in Dexing area exceeded the permit level, and that crop types and Cd concentrations in soils have a substantial effect on the uptake and accumulation of Cd in crops. Ni et al. [27,28] studied the HMs content in scalp hair of the inhabitants near Dexing Copper Mine and found that The HMs levels in the scalp hair were closely related to local geological environment, and that the Pb and Cd were highest in the child group compared to other groups. However, few research has been conducted on the environmental risks of HMs caused by different types of deposit in the study area, and the systematic studies on HMs contamination in surface water, ground water, and sediments are lacking [29–33].

Therefore, the major objective of this study is to fill this gap. Hence, the total concentrations of HMs (As, Cd, Cu, Cr, Hg, Pb, and Zn) and pH values of the surface water, ground water, and sediments were estimated to evaluate the pollution and environmental risk using pollution indices ($P_i$ and $P_n$), geo-accumulation index ($I_{geo}$), and potential ecological risk index ($RI$) in the Dexing area. The novelty of this study is the systematic collecting of three types of environmental medium samples, the comparing of the pollution characteristics of HMs in the three main types of deposits in Dexing area, and the comprehensive evaluation of the ecological risk of HMs. Accordingly, we (1) examined the concentrations of HMs in the surface/ground waters and sediments around the Dexing ore cluster, (2) compared the spatial distribution and combined the characteristics of HM contamination in the study area, and (3) assessed the contamination levels and potential ecological risk of HMs.

## 2. Materials and Methods

### 2.1. Study Area

The Dexing ore cluster is one of the most important Cu–Au (Pb–Zn–Ag) production areas in China, where the proven reserves of Cu and Au are >11 million tons and 600 tons, respectively [22]. The Dexing porphyry copper mine (primarily comprising Tongchang, Fujiawu, and Zhushahong deposits), Yinshan Pb–Zn polymetallic mine, and Jinshan gold mine are the major mines in the study area [34–36]. After more than 50 years of large-scale mining, many mining pits, tailing ponds, mine dumps, heap leaching fields, and

ore-dressing plants have been established in the area. The Dexing area features the largest copper mine open pit (Tongchang mining pit) and tailing pond (4# tailing pond) in Asia, as shown in Figure 1.

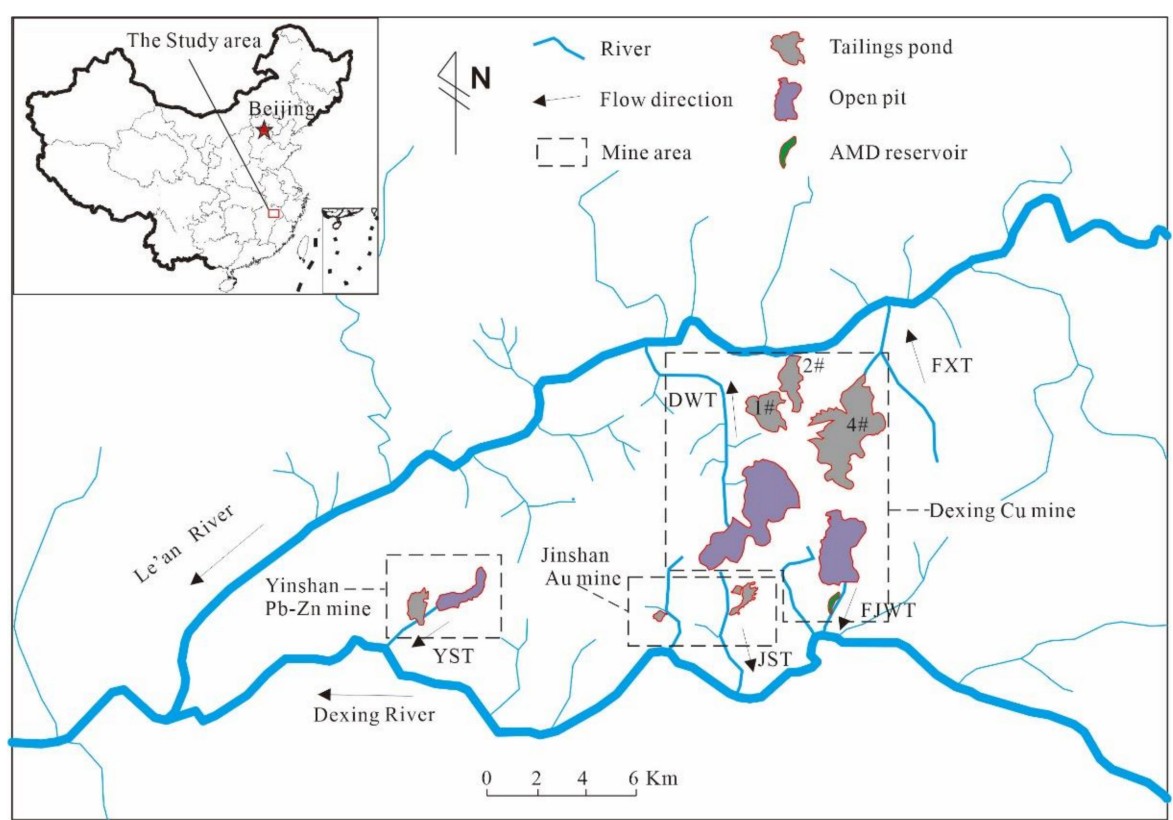

**Figure 1.** Location of the study area. FXT-Fuxi tributary, DWT-Dawu tributary, FJWT-Fujiawu tributary, JST-Jinshan tributary, YST-Yinshan tributary.

The study area is in the middle subtropical zone with prevalent rainy and humid climate, a hot summer, and a warm winter. The average annual temperature and annual mean precipitation are 17.2 °C and 1853 mm, respectively. The Le'an River (north) and the Dexing River (south) (Figure 1) are the major rivers flowing through the study area. The Le'an River is one of tributaries of the Poyang Lake, which represents the largest freshwater lake in China. The Le'an River has attracted considerable attention from environmentalists owing to its severe regional HMs pollution [30,33,37,38].

The Fuxi and Dawu are the tributaries of the Le'an River. The Le'an River flows through 2# tailing pond of Dexing copper mine, Fuxi tributary flows through 4# tailing pond, and Dawu tributary flows through the 1# tailing pond, open pit, mine dump, heap leaching field, and ore-dressing plants of Tongchang Cu deposit. The Fujiawu, Jinshan, and Yinshan are the tributaries of the Dexing River. The Fujiawu tributary flows through the open pit, AMD reservoir, and ore-dressing plants of Fujiawu Cu deposit, whereas the Jinshan tributary flows through the mining pit, tailing ponds, and ore-dressing plants of Jinshan Au mine. The Yinshan tributary flows through the mining pit, tailing ponds, mine dump, and ore-dressing plants of Yinshan Pb–Zn polymetallic mine.

*2.2. Sampling*

Water samples were collected using 250 mL polyethylene plastic bottles, which were rinsed 3–4 times using the in-situ water before the samples were obtained. Briefly, 114 surface water samples were collected from the Le'an River, Dexing River, and their tributaries. Twenty-seven ground water samples were collected from residential wells. All the water samples were filtered using paper filters to remove suspended particles. Afterward,

a drop of $HNO_3$ (50%) was added to make their pH < 2 for preservation. Subsequently, the pH of surface and ground water was measured using a portable multi-parameter instrument (ORION 4 STAR, Thermo Fisher, Waltham, MA, USA) during the sampling. In total, 200 superficial sediment samples (0–10 cm) were collected through multi-point mixing method [39]. Afterward, they were stored in clean polyethylene bags, and each sample was air dried and subsequently sieved through a 2 mm mesh to remove plant fragments and stones.

*2.3. Analytical Methods*

2.3.1. Analysis of Surface and Ground Water Samples

After filtration using 0.45 μm membrane filter, the concentrations of Cd, Cr, Cu, Pb, and Zn in water samples were analyzed through inductively coupled plasma-mass spectrometry (ICP-MS; XSERIES 2, Thermo Fisher, USA). Briefly, 20 mL from each water sample was obtained, and 5 mL aqua regia (1:1) was added. Afterward, As and Hg concentrations were analyzed via the atomic fluorescence spectrometry (AFS; 8130, Jitian Instruments, China). The standard samples (GSB04-1767-2004 and GSB04-1729-2004) were used for quality control, and 29 parallel samples were analyzed. The results show that the precision (RED) of all elements was <15%, and the relative deviation (RE) of parallel sample was <20%.

2.3.2. Analysis of Sediment Samples

During the analysis of the sediment samples, pH was measured on the 2 mm fraction in a 1:2.5 sediment/water (*m/v*) solution using a pH meter (PHS-3C, Rex Instruments, China) after shaking for 30 min. The sieved sediment samples (<2 mm) were ground to fine power (<0.074 mm) by agate grinder for determining Ag, As, Cd, Cu, Cr, Hg, Pb and Zn. The concentration of As and Hg were quantified by AFS (8130, Jitian Instruments, Beijing, China), Cd and Cu concentrations were quantified using ICP-MS (XSERIES 2, Thermo Fisher, Waltham, MA, USA), and the concentrations of Cr, Pb, Zn, and S were analyzed via X-ray fluorescence spectrometry (XRF; PW4400/40, PANalytical, Almelo, The Netherlands). The methods used have been provided by Cheng et al. (2014) [40]. Four certified reference materials (GBW 07408, GBW 07424, GBW 07427, and GBW 07429) and two duplicate samples were inserted and analyzed for each batch of 50 samples. The RSD were <10% for all elements, and the RE of duplicate sample was <15%.

2.3.3. Statistical Analysis

Prior to further analysis, the data of HMs were centered log-ratio (clr) transformed to identify anomalous multi-element associations [40,41]. Furthermore, statistical analyses were performed using the PASW Statistic 18.0 software (IBM, Chicago, IL, USA). Hence, cluster analysis (CA) was conducted for analyzing the relationships and combined characteristics of HMs using Ward's method with Euclidean distances as a measure of similarity [12].

*2.4. Geochemical and Environmental Risk Assessment*

2.4.1. Pollution Indices ($P_i$ and $P_n$)

The pollution index ($P_i$) and the Nemerow integrated pollution index ($P_n$) were applied to assess the pollution of HMs in water according to Equations (1) and (2) as shown below [8]:

$$P_i = \frac{C_i}{S_i} \tag{1}$$

$$P_n = \sqrt{\frac{I_{Avg}^2 + I_{Max}^2}{2}} \tag{2}$$

where $C_i$ is the concentration of metal *i* in samples; $S_i$ is its reference concentrations, which are Chinese Quality Criterion for Surface Water (class III) [42] for surface water samples

and Standards for Chinese Drinking Water Quality [43] for ground water samples; $I_{Avg}$ is the mean value of all $P_i$ of the HMs considered; and $I_{Max}$ is the maximum value.

### 2.4.2. Geo-Accumulation Index ($I_{geo}$)

The geo-accumulation index evaluates the enrichment of HM levels above background values. It has been widely used to assess HM contamination in sediments [12] and can be calculated using Equation (3):

$$I_{geo} = log_2 \left[ \frac{C_i}{1.5 \times B_i} \right] \tag{3}$$

where $C_i$ is the concentration of metal $i$ in samples, and $B_i$ is the geochemical background value of stream sediments in southern China [39].

### 2.4.3. Potential Ecological Risk Index ($RI$)

As $I_{geo}$ does not consider the toxicity and the combined effects of each HMs, $RI$ was used to estimate the potential ecological risk of HMs to the aquatic ecosystem [44]. The $RI$ is calculated using Equations (4)–(6):

$$C_f^i = \frac{C_i}{B_i} \tag{4}$$

$$E_{ir} = C_f^i \times T_r^i \tag{5}$$

$$RI = \sum E_{ir} \tag{6}$$

where $C_f^i$ and $E_{ir}$ represent the contamination factor and ecological risk of each HMs, respectively; the values of $C_i$ and $B_i$ are the same as Equation (3); $T_r^i$ is the toxic response factor, which for As, Cd, Cr, Cu, Hg, Pb, and Zn are 10, 30, 2, 5, 40, 5 and 1, respectively [44]; and $RI$ is the sum of the $E_{ir}$ of each HMs. The classes of $P_i$, $P_n$, $I_{geo}$, and $RI$ are listed in Table 1.

**Table 1.** Evaluation criteria of pollution indices ($P_i$ and $P_n$), geo-accumulation index ($I_{geo}$), and risk index ($E_{ir}$ and $RI$).

| $P_i$ Class [45] | Pollution Level | $P_n$ Class [38] | Pollution Level | $I_{geo}$ Class [46] | Pollution Level | $E_{ir}$ Class | Potential Risk | $RI$ Class [44] | Ecological Risk |
|---|---|---|---|---|---|---|---|---|---|
| <1 | Safe | <0.7 | Safe | <0 | Unpolluted | <40 | Low | <150 | Low |
| 1–2 | Slight | 0.7–1 | Precaution | 0–1 | Unpolluted to Moderate | 40–80 | Moderate | 150–300 | Moderate |
| 2–3 | Moderate | 1–2 | Slight | 1–2 | Moderate | 80–160 | High | 300–600 | High |
| 3–5 | Heavy | 2–3 | Moderate | 2–3 | Moderate to Heavy | 160–320 | serious | >600 | serious |
| >5 | Extreme | >3 | Heavy | 3–4 | Heavy | >320 | severe | | |
| | | | | 4–5 | Heavy to Extreme | | | | |
| | | | | >5 | Extreme | | | | |

## 3. Results and Discussion

### 3.1. HMs Concentrations

3.1.1. Surface and Ground Water

The HMs concentrations and pH in the surface/ground water samples and permissible values for surface water (GB 3838-2002 class III) and drinking water (GB.5749-2006) are presented in Figure 2 and Table 2. The pH and HMs concentrations of an aquatic system

represent its water quality. The concentrations of HMs and pH value in the study area highly varied, indicating that the surface water was severely disturbed by mining activities. The pH value of surface water varied in the range of 2.40–8.50, with an average of 6.26. The Fujiawu tributary had the lowest pH of surface water (ranging 2.40–4.24), followed by Yinshan and Dawu tributaries, with variations in the range of 2.79–4.40 and 2.47–7.22, respectively. The pH values of other streams were close to neutral (ranging 6.00–8.50). The surface water acidification might be caused by the AMD, particularly, due to the mining pit, mine dump, heap leaching field, AMD reservoir, and ore-dressing plants of Cu–Pb–Zn(Ag) sulfide deposits, for example, in the Dexing porphyry Cu mine and Yinshan Pb–Zn(Ag) polymetallic mine. Compared with other metal mining areas, the surface water in Dexing area had lower pH values, indicating that it has been more severely affected by AMD [8,12,19,45]. However, we argue that the acidification of the surface water downstream of the #4 tailing pond and Jinshan Au mine was not prominent. AMD is an important carrier of HMs [46,47]. The mean concentrations of HMs in the surface water followed the order Zn > Cu > Cd > As > Pb > Cr > Hg, with the values (mean ± standard deviation, e.g., SD) of 2529 ± 11,791, 2405 ± 13,396, 213 ± 1261, 71.3 ± 457, 52.8 ± 280, 18.5 ± 73.8, and 0.0372 ± 0.0229 µg/L, respectively. All the HM concentrations, except Hg, significantly varied among the rivers and tributaries. Furthermore, As, Cd, Cr, and Cu concentrations were highest in Fujiawu tributary (945 ± 1557, 2859 ± 4124, 143 ± 180, and 14,325 ± 32,074 µg/L, respectively), whereas Pb and Zn concentrations were highest in the Yinshan tributary (570 ± 600 and 45,829 ± 37,022 µg/L, respectively). Our results indicate that the acidification of surface water is associated with the HMs concentrations, thereby suggesting that the HMs pollution around the Dexing area could be severely affected by AMD. The pH values of the Le'an and Dexing rivers were ~7, and the HMs concentrations were relatively low. This finding indicates that the major rivers had significant dilution effect on the surface water from the tributaries.

The groundwater in the study area was slightly acidified, and the pH value ranged from 5.39 to 7.31, with an average of 6.19. The mean concentration of HMs in groundwater followed the order of Zn > Cu > Cr > As > Pb > Cd > Hg and ranged between 2.76–168, 0.3–27.9, 1.62–5.13, 0.06–12.0, 0.2–19.2, 0.013–0.42, and 0.0104–0.0497 µg/L, respectively. Except the maximum value of As and Pb, the concentrations of the HMs in groundwater were all lower than the threshold value of drinking water.

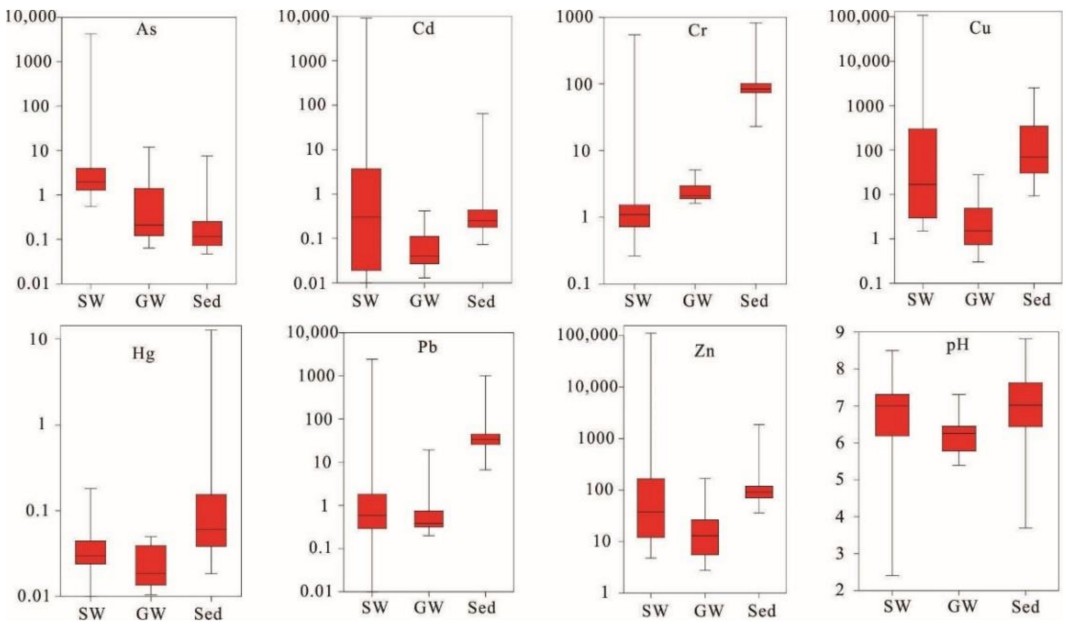

**Figure 2.** HM concentrations in the surface water (SW, n = 114), ground water (GW, n = 27), and sediment (Sed, n = 200) samples (unit is µg/L for water and µg/g for sediment samples).

**Table 2.** Summary of the guideline values and statistics of HMs and pH in the surface and ground water samples.

| Location | | As | Cd | Cr | Cu | Hg | Pb | Zn | pH |
|---|---|---|---|---|---|---|---|---|---|
| SW | | | | | | | | | |
| FXT | Mean | 1.89 ± 0.81 | 0.26 ± 0.11 | 1.12 ± 0.25 | 4.64 ± 3.07 | 0.024 ± 0.010 | 0.62 ± 0.78 | 35.8 ± 39.5 | 7.15 ± 0.24 |
| n = 11 | Range | 0.81~3.48 | 0.16~0.48 | 0.77~1.56 | 1.51~10.5 | 0.005~0.040 | 0.08~2.71 | 5.22~101 | 6.80~7.50 |
| DWT | Mean | 1.76 ± 1.94 | 2.64 ± 7.58 | 13.1 ± 49.9 | 5689 ± 22,503 | 0.038 ± 0.019 | 1.38 ± 1.45 | 507 ± 1792 | 5.26 ± 1.56 |
| n = 23 | Range | 0.55~10.1 | 0.009~37.2 | 0.34~241 | 2.53~108,700 | 0.009~0.073 | 0.01~5.7 | 7.06~8715 | 2.47~7.22 |
| LAR | Mean | 1.36 ± 0.33 | 0.13 ± 0.15 | 1.04 ± 0.4 | 32.1 ± 80.7 | 0.043 ± 0.023 | 0.59 ± 0.26 | 27.1 ± 24.6 | 7.04 ± 0.27 |
| n = 19 | Range | 0.74~2.29 | 0.01~0.54 | 0.52~2.36 | 1.86~362 | 0.012~0.108 | 0.29~1.11 | 4.85~81.2 | 6.41~7.55 |
| FJWT | Mean | 945 ± 1557 | 2859 ± 4124 | 143 ± 180 | 14325 ± 32,074 | 0.045 ± 0.027 | 382 ± 840 | 4732 ± 5100 | 3.00 ± 0.60 |
| n = 8 | Range | 1.03~4275 | 0.67~9107 | 0.76~549 | 126~93,281 | 0.016~0.090 | 1.72~2435 | 82.7~12,460 | 2.40~4.24 |
| JST | Mean | 19.7 ± 9.43 | 0.012 ± 0.005 | 1.26 ± 0.2 | 2.2 ± 0.33 | 0.035 ± 0.014 | 0.1 ± 0.1 | 23.1 ± 20.1 | 7.24 ± 0.23 |
| n = 8 | Range | 2.15~30.7 | 0.01~0.02 | 1.09~1.72 | 1.67~2.59 | 0.022~0.064 | 0.01~0.27 | 5.89~56.7 | 7.01~7.68 |
| YST | Mean | 30.7 ± 35 | 189 ± 141 | 117 ± 185 | 5291 ± 4152 | 0.070 ± 0.066 | 570 ± 600 | 45829 ± 37,022 | 3.27 ± 0.67 |
| n = 5 | Range | 10.3~92.6 | 120~442 | 1.35~442 | 1606~12,010 | 0.017~0.182 | 78~1611 | 24420~111,700 | 2.79~4.40 |
| DXR | Mean | 4.0 ± 5.8 | 13.5 ± 43.1 | 0.96 ± 0.38 | 22.2 ± 15.4 | 0.035 ± 0.017 | 1.47 ± 1.82 | 302 ± 814 | 7.20 ± 0.56 |
| n = 24 | Range | 0.74~30.6 | 0.014~215 | 0.46~1.73 | 2.61~59.4 | 0.017~0.081 | 0.23~5.71 | 4.77~3914 | 6.00~8.50 |
| SW-All | Mean | 71.3 ± 457 | 213 ± 1261 | 18.5 ± 73.8 | 2405 ± 13,396 | 0.037 ± 0.023 | 52.8 ± 280 | 2529 ± 11,791 | 6.26 ± 1.63 |
| n = 114 | Range | 0.55~4275 | 0.009~9107 | 0.26~549 | 1.51~108,700 | 0.005~0.182 | 0.01~2435 | 4.77~111,700 | 2.40~8.50 |
| GB 3838-2002 Class III | | 50 | 5 | 50 | 1000 | 0.1 | 50 | 1000 | 6.0~9.0 |
| GW | Mean | 1.3 ± 2.48 | 0.08 ± 0.09 | 2.54 ± 0.97 | 3.43 ± 5.44 | 0.024 ± 0.013 | 1.24 ± 3.61 | 23.8 ± 34.9 | 6.19 ± 0.48 |
| n = 27 | Range | 0.06~12 | 0.013~0.42 | 1.62~5. 13 | 0.3~27.9 | 0.010~0.05 | 0.2~19.2 | 2.76~168 | 5.39~7.31 |
| GB 5749-2006 | | 10 | 5 | 50 | 1000 | 1 | 10 | 1000 | 6.5~8.5 |

Unit is μg/L for HMs; n-number, FXT-Fuxi tributary, DWT-Dawu tributary, LAR- Le'an River, FJWT-Fujiawu tributary, JST-Jinshan tributary, YST-Yinshan tributary, DXR-Dexing River.

### 3.1.2. Sediments

The statistics of HMs concentrations, pH, and the background value of sediments are listed in Table 3. Due to the challenges during the sampling process, there was only one sample in the Yinshan tributary, which was close to the tailing pond of Yinshan Pb–Zn (Ag) polymetallic mine. Hence, the HM concentration and pH estimate of the Yinshan tributary is the only reference that is not used in the following discussion, and this limitation has not hampered the conclusions.

**Table 3.** Summary of the background values and basic statistics of HMs and pH in the sediment samples.

| Location | | As | Cd | Cr | Cu | Hg | Pb | Zn | pH |
|---|---|---|---|---|---|---|---|---|---|
| FXT | Mean | 14.8 ± 3.82 | 0.17 ± 0.06 | 93.3 ± 17.5 | 415 ± 464 | 0.046 ± 0.013 | 21.3 ± 10.7 | 73.1 ± 29.3 | 7.72 ± 0.65 |
| n = 11 | Range | 6.89~21.9 | 0.08~0.26 | 71~122 | 42.9~1410 | 0.026~0.071 | 6.6~35.5 | 36~108 | 6.89~8.82 |
| DWT | Mean | 35.3 ± 35.6 | 0.3 ± 0.15 | 102 ± 27 | 876 ± 743 | 0.61 ± 2.31 | 52.2 ± 28 | 107 ± 41.6 | 5.83 ± 1.58 |
| n = 30 | Range | 6.37~190 | 0.08~0.6 | 69~203 | 39.2~2538 | 0.025~12.77 | 9.2~155 | 38~217 | 3.69~8.43 |
| LAR | Mean | 17.7 ± 35.7 | 1.99 ± 10.8 | 85.9 ± 13.9 | 184 ± 189 | 0.093 ± 0.16 | 52.6 ± 151 | 81.1 ± 49.7 | 6.81 ± 0.94 |
| n = 36 | Range | 4.58~220 | 0.07~65.1 | 68~144 | 16~680 | 0.018~0.79 | 17~930 | 50~280 | 4.18~8.52 |
| FJWT | Mean | 289 ± 414 | 0.68 ± 0.62 | 115 ± 10.5 | 370 ± 221 | 0.085 ± 0.044 | 99.3 ± 72 | 123 ± 53.7 | 4.75 ± 0.87 |
| n = 12 | Range | 9.36~1428 | 0.09~1.71 | 94~126 | 55.7~860 | 0.028~0.15 | 20.3~235 | 50~194 | 3.75~6.8 |
| JST | Mean | 509 ± 199 | 0.3 ± 0.12 | 93.9 ± 7.22 | 50.2 ± 21.2 | 1.04 ± 0.45 | 33 ± 8.85 | 124 ± 20.7 | 8.01 ± 0.35 |
| n = 8 | Range | 202~820 | 0.17~0.47 | 84~106 | 26.9~89.8 | 0.30~1.62 | 19.3~42.7 | 99~149 | 7.6~8.5 |
| YST | n = 1 | 893 | 5.22 | 145 | 1369 | 0.36 | 1000 | 1883 | 7.88 |
| DXR | Mean | 98.7 ± 122 | 1.25 ± 1.79 | 70 ± 22.7 | 254 ± 216 | 0.094 ± 0.082 | 65.2 ± 66.5 | 151 ± 83.5 | 7.42 ± 0.57 |
| n = 29 | Range | 12.2~600 | 0.23~7.91 | 34~123 | 13.8~1035 | 0.025~0.34 | 26.2~253 | 82~382 | 5.99~8.17 |
| All | Mean | 75.8 ± 172 | 0.8 ± 4.65 | 93.2 ± 60.3 | 279 ± 441 | 0.26 ± 0.95 | 53.1 ± 99.9 | 117 ± 143 | 6.81 ± 1.19 |
| n = 200 | Range | 0.88~1428 | 0.07~65.1 | 23~823 | 9.26~2538 | 0.018~12.8 | 6.6~1000 | 36~1883 | 3.69~8.82 |
| Background value | | 13.1 | 0.23 | 67 | 25 | 0.075 | 32.2 | 81 | |

Unit is μg/g for HMs.

The pH values of the sediment samples ranged from 3.69 to 8.82 with a mean value of 6.81. The sediments in Fujiawu tributary exhibited the lowest pH (3.75–6.8), followed by Dawu tributary (3.69–8.43). The sediments with low pH, in which the HMs are generally more mobile, might be impacted by hydrogen ions of AMD. The sediments in the Fuxi, Jinshan, and Yinshan tributaries were primarily weakly alkaline, indicating that copper tailing ponds and gold mining activities in the study area were not the major sources of AMD.

The HMs concentrations in sediments also varied significantly and followed the order of Cu > Zn > Cr > As > Pb > Cd > Hg, ranging 9.26–2538, 36–1883, 23–823, 0.88–1428, 6.6–1000, 0.07–65.1, and 18.4–12,771 µg/g, respectively. The mean concentrations of HMs were significantly higher than the background values; that is, the mean concentrations of As, Cd, Cu, and Hg were 5.8, 3.5, 11.2, and 3.4 times greater than their background values, respectively. Furthermore, the Cu mean concentration was highest in Dawu tributary (876 µg/g), Cr and Pb mean concentrations were highest in Fujiawu tributary (115 and 99.3 µg/g, respectively), As and Hg mean concentrations were highest in Jinshan tributary (509 and 1.04 µg/g, respectively), Cd mean concentration was highest in Le'an River (1.99 µg/g), and Zn mean concentration was highest in Dexing River (151 µg/g).

*3.2. Spatial Distribution and Combined Characteristics of HMs*

3.2.1. Spatial Distribution of HMs

As shown in Figure 3, the distribution of HMs in the ground water exhibited substantial spatial heterogeneity. The narrow ranges of Cr and Hg contents in the ground water indicated minor exposure to the mining activities. The highest values of Cu and As were clustered in the downstream of the 2# tailing pond of the copper mine and of Jinshan Au mine, respectively. We revealed similar distribution patterns for Pb, Zn, and Cd, with the highest values in downstream of Yinshan Pb–Zn polymetallic mine.

The distributions of HMs in the surface water are shown in Figure 4. Notably, Cu, Cr, Cd, Pb, and Zn exhibited similar spatial patterns. Their hotspots were clustered at the tributaries of the Cu and Pb–Zn polymetallic mines, whereas the contents in major rivers decreased rapidly. The distribution pattern of As was slightly different from the above elements, with moderate–high content at Au mine. Mercury exhibited a different distribution pattern, whose hotspots were not only clustered in tributaries, but also in major rivers. Although flowing through one of the largest tailings ponds in Asia (4# pond), the HMs concentration in surface water of Fuxi tributary did not increase significantly. By Combining the study of Azhari et al. [12], it can be concluded that the tailings pond was not the main source of HMs in surface water of metal mine areas.

Figure 5 also shows that As, Pb, Zn, and Cd contents in the sediment samples had spatial similarities with those in the surface water. Interestingly, the sediment sample with the maximum Pb and Cd content was in the industrial park which exhibited the largest lead smelting enterprise in Jiangxi Province, which was consistent with the results of Teng et al. [26]. Copper, Cr, and Hg in the sediments exhibited unique spatial distribution patterns. The high concentrations of Cu appeared in the downstream of the Tongchang Cu deposit, which is the oldest deposit in the study area. The high concentration samples were downstream from the Au mine for Hg. Furthermore, spatial variations of the Cr concentrations were relatively low, and high concentrations were clustered primarily around Cu deposits. High HMs concentrations in surface water were primarily clustered at the tributaries closest to the mine areas, and the sediments were extended several kilometers downstream of the major rivers.

3.2.2. Cluster Analysis

CA was applied to the group of HMs with homologous characteristics. In the ground water, three distinct clusters can be outlined as shown in Figure 6a. Cluster 1 contained Cr and Hg; these HMs exhibited lower concentrations compared with the corresponding guidelines. Moreover, they exhibited the lowest spatial variabilities, compared to other

HMs, as presented in Table 2. This finding suggests that the elements mostly originated from the natural sources. Cluster 2 contained Pb, Cd, Cu, and Zn, and the concentrations of these elements were below their guidelines, except Pb in one sample. They also had large spatial variations, compared with cluster 1. Arsenic as the third group, exhibited the greatest spatial variability, and one sample exceeded the corresponding guideline threshold. This finding indicates that the HMs in cluster 2 and 3 originated from both natural and anthropogenic sources.

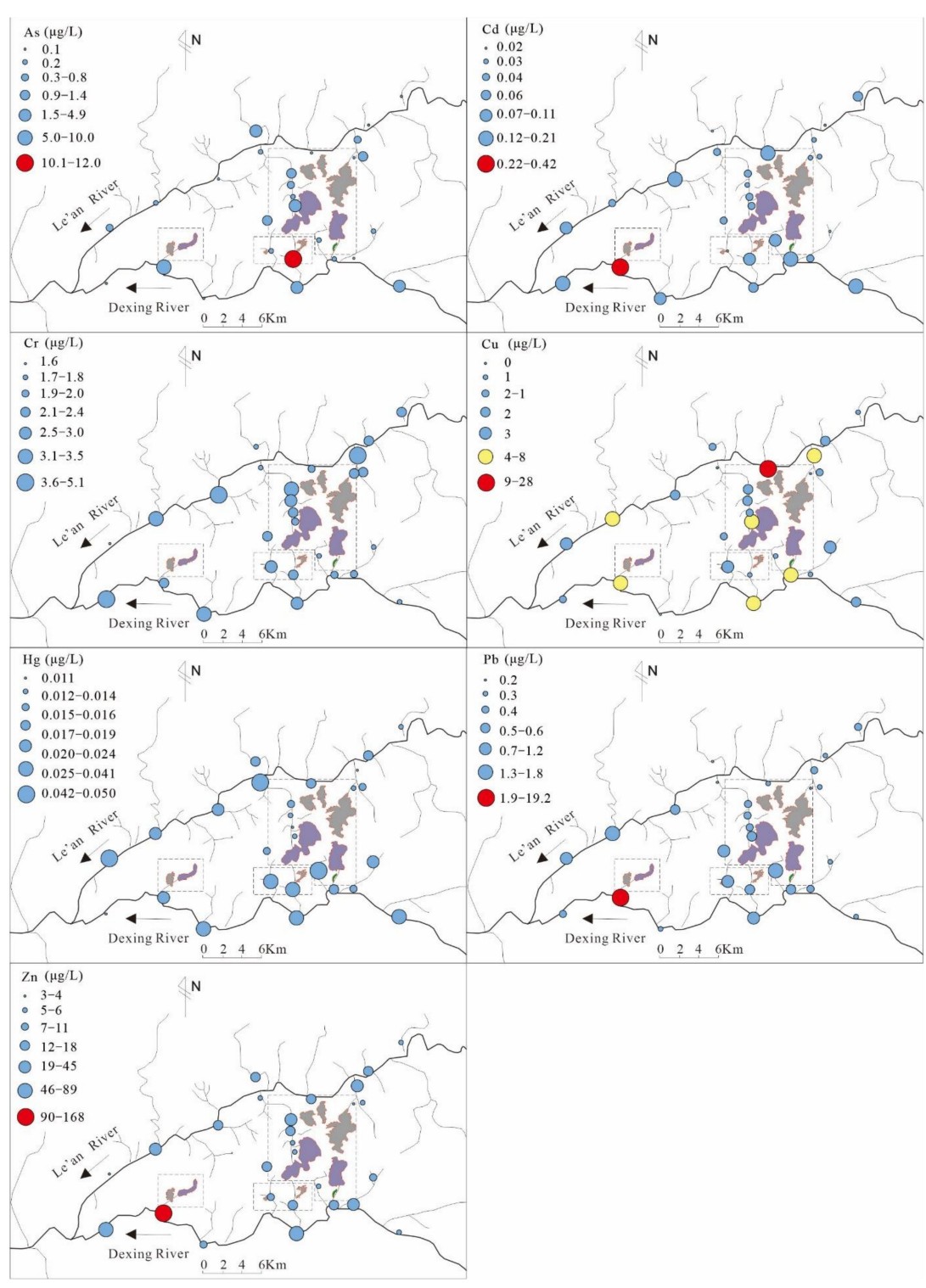

**Figure 3.** HMs distribution in ground water samples.

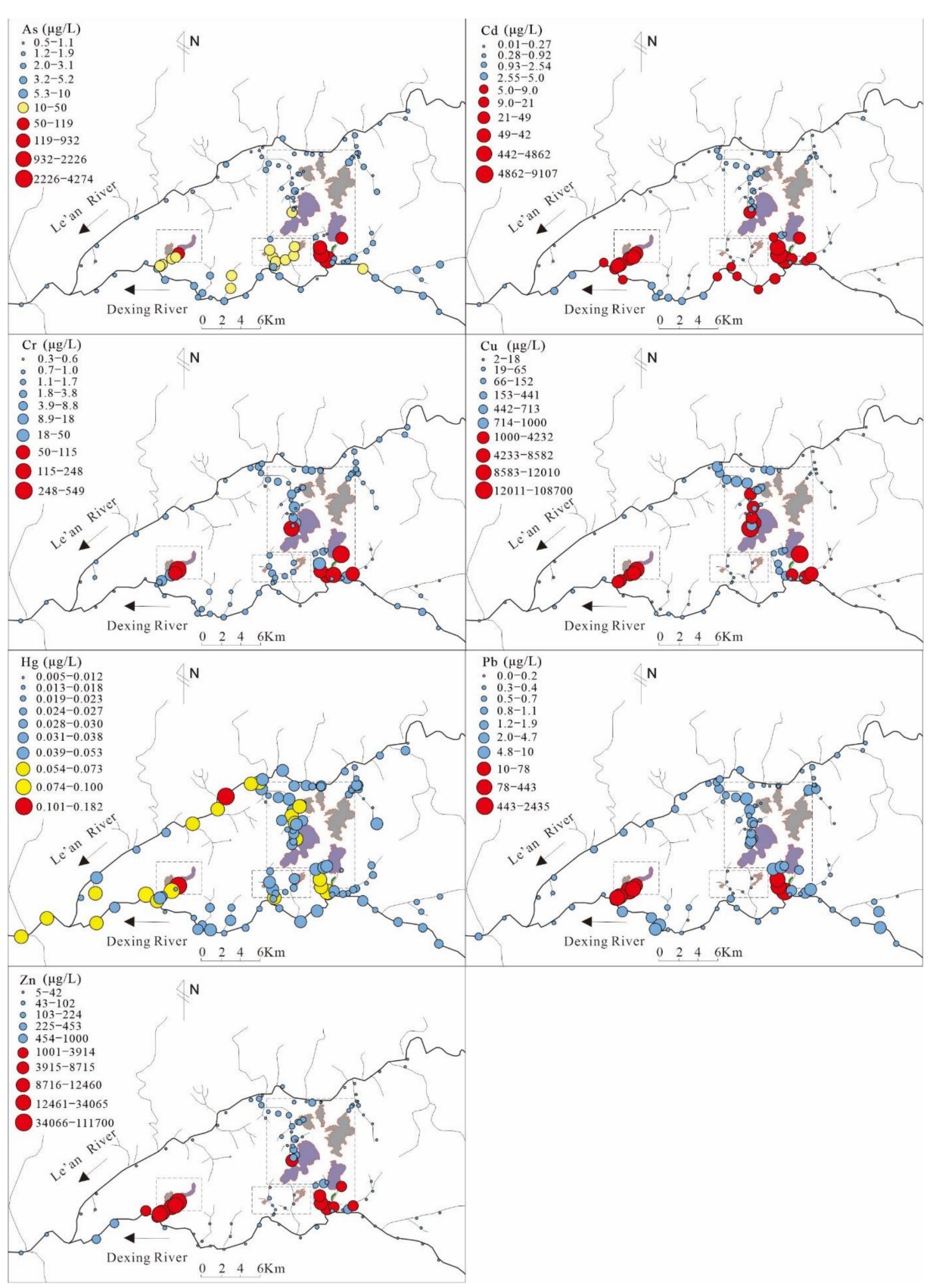

**Figure 4.** HMs distribution in surface water samples.

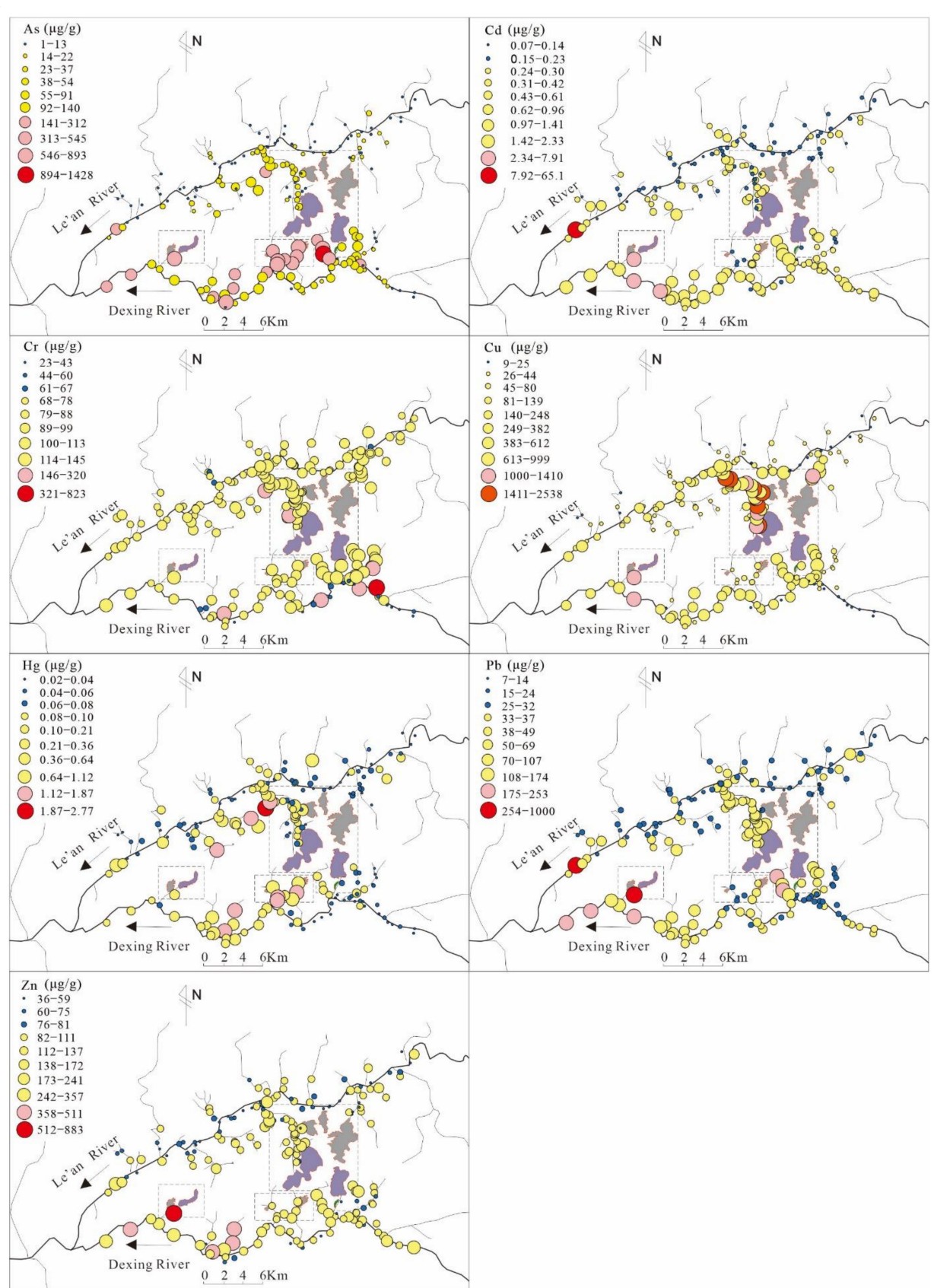

**Figure 5.** HMs distribution in sediments samples.

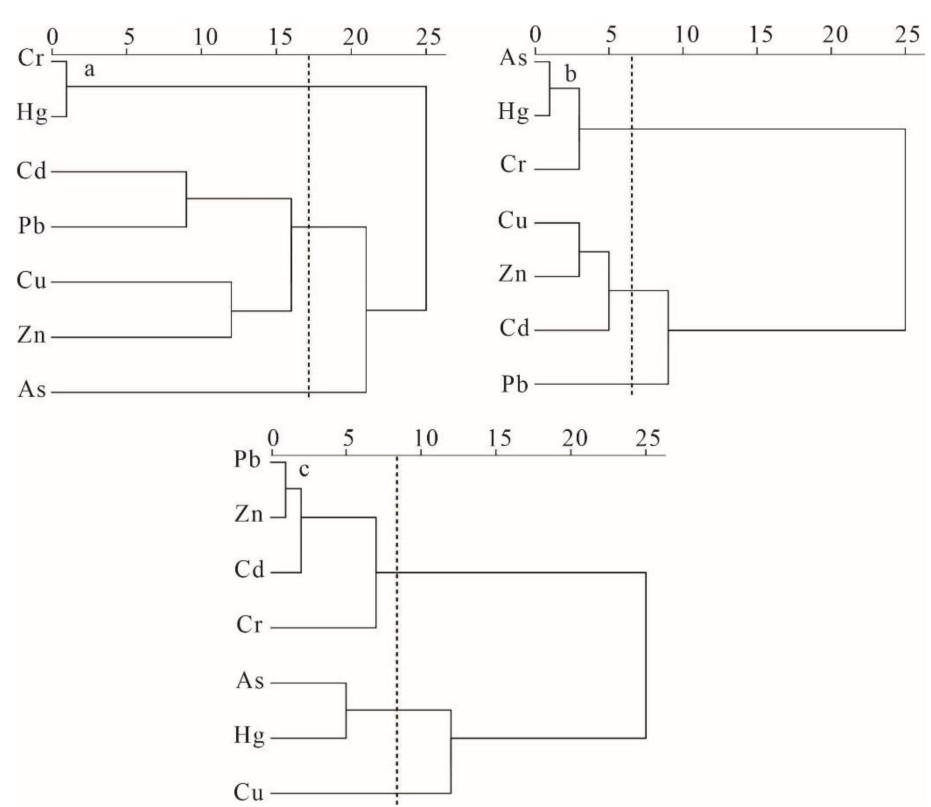

**Figure 6.** Dendrogram showing the clustering of the HMs in (**a**) ground water, (**b**) surface water, and (**c**) sediment samples.

In the surface water, HMs can be grouped into three clusters, as illustrated in Figure 6b. The first group included Cu, Zn and Cd, first two of which were the major mineralizing elements in the study area, and the last had similar geochemical characteristics to Zn. Their high concentrations and strong spatial variability suggest that Cu, Zn, and Cd primarily originated from the anthropogenic sources (Table 2). As for the second group, Pb was also one of major mineralizing elements; however, it exhibited low concentration and spatial variability, compared with Cu, Zn, and Cd owing to its weak migration ability in surface water [48]. Cluster 3 contained As, Hg, and Cr. Due to low concentrations, Hg and Cr may originated from the natural sources; however, the mean content of As was above its guideline threshold. Thus, it may be affected by the anthropogenic activities.

In the sediments, HMs were grouped into three different clusters, which were consistent with the 3 major types of deposits (copper, gold and lead-zinc mines) in the study area (Figure 6c). Cluster 1 contained Cu, that was extremely enriched in the sediments. The mean content was more than ten-fold compared with the background value, as observed in Table 3. The second group comprised As and Hg, which was primarily associated with the gold mines [7], and the mean concentrations were 5.7 and 3.4 times higher than their background values, respectively. Cluster 3 contained Pb, Zn, and Cd, and the mean concentrations were 1.4–3.4 times higher than their background values. Owing to high concentrations and spatial variability, all HMs primarily originated from the anthropogenic sources.

The spatial distribution and CA of HMs in the study area indicate the following pattern. The degree of impact on different environmental media by anthropogenic activities (e.g., mining, beneficiation, and smelting) followed the order of sediment > surface water > ground water. During the mining and smelting in the Dexing area, the oxidation of sulfides such as pyrite, chalcopyrite, sphalerite, galenite, and arsenopyrite introduced high concentrations of HMs and sulfate ions to waters, thereby generating AMD. Moreover, through the surface runoff and aerial transport, mine tailings, waste waters, and dusts could release a substantial number of HMs to the surrounding environment. The sediment is a carrier of HMs in

rivers; thus, HMs are easy to be absorbed in particles. Hence, it causes the transportation and accumulation of HMs in sediments. The HMs concentrations in sediments therefore can reflect the degree of HMs pollution [20]. The solubility of HMs in water is generally low, particularly in neutral and alkaline environment. In our analysis, the HMs concentrations in water were closely related to the pH value. Owing to the generation of AMD, the HMs concentrations in surface water near the deposits were extremely high but decreased rapidly as the distance increased. Owing to the obstruction and purification of rocks, the groundwater in the Dexing area was less affected by anthropogenic activities.

### 3.3. Pollution Characteristics and Environmental Risk Assessment

3.3.1. Ground and Surface Water

Tables 4 and 5 present the number of different $P_i$ and $P_n$ levels of the HMs in the ground water and surface water, respectively. Briefly, 100% samples of Cd, Cr, Cu, Hg, and Zn in ground water were not polluted. Both As and Pb had one sample classified into the slightly polluted group. The $P_n$ ranged from 0.03 to 1.38. Based on the category of $P_n$, one sample was slightly polluted, and one sample was in the group of precaution in ground water.

**Table 4.** Number of different $P_i$ and $P_n$ levels for ground water samples (n = 27).

| Pollution Level | $P_i$ | | | | | | | Pollution Level | $P_n$ |
| --- | --- | --- | --- | --- | --- | --- | --- | --- | --- |
| | As | Cd | Cr | Cu | Hg | Pb | Zn | | |
| Safe | 26 | 27 | 27 | 27 | 27 | 26 | 27 | Safe | 25 |
| Slight | 1 | 0 | 0 | 0 | 0 | 1 | 0 | Precaution | 1 |
| | | | | | | | | Slight | 1 |

**Table 5.** Number of different $P_i$ and $P_n$ levels for surface water samples (n = 114).

| Pollution Level | $P_i$ | | | | | | | Pollution Level | $P_n$ |
| --- | --- | --- | --- | --- | --- | --- | --- | --- | --- |
| | As | Cd | Cr | Cu | Hg | Pb | Zn | | |
| Safe | 109 | 91 | 106 | 100 | 112 | 106 | 100 | Safe | 86 |
| Slight | 1 | 8 | 3 | 1 | 2 | 1 | 4 | Precaution | 4 |
| Moderate | 1 | 2 | 1 | 2 | 0 | 0 | 0 | Slight | 5 |
| Heavy | 0 | 2 | 2 | 4 | 0 | 0 | 1 | Moderate | 2 |
| Extreme | 3 | 11 | 2 | 7 | 0 | 7 | 9 | Heavy | 17 |

The pollution level of the surface water was higher than that of the ground water. The proportions of moderate–extreme polluted samples for each HMs were in the following order: Cd (13.2%) > Cu (11.4%) > Zn (8.8%) > Pb (6.1%) > Cr (4.4%) > As (3.5%) > Hg (0%). The numbers of samples of Cd, Zn, Cu, Pb, As, and Cr that reached the level of extreme pollution were 11, 9, 7, 7, 3, 2, respectively. The $P_n$ ranged from 0.04 to 1301. Based on the category of $P_n$, four samples were slightly polluted, five samples were in the group of precaution, two samples were moderate polluted, and seventeen samples were heavily polluted.

Furthermore, Figure 7 shows that surface water in different streams had different types and degrees of pollution. Notably, copper was the major pollutant in Dawu tributary (21.7% samples reached moderate to extreme level). In Fujiawu tributary, the proportions of the surface water reached moderate to extreme pollution level with 75% for Cd, 50% for Cu, 50% for As, 37.5% for Pb, 37.5% for Zn, and 37.5% for Cr. In Yinshan tributary, Cd and Zn in all the samples were extremely polluted; 83.3% of the samples reached extreme level for Pb, and 83.3% and 33.3% of the samples reached moderate to extreme level for Cu and Cr, respectively. In contrast, the pollution level of the surface water in the major streams were markedly lower. There was only one sample (5.3%) for Hg classified into slightly polluted in Le'an River. In Dexing River, 12.5% and 4.2% of samples reached moderate to

extreme level for Cd and Zn, respectively. All the samples from Jinshan tributary and Fuxi tributary were safe.

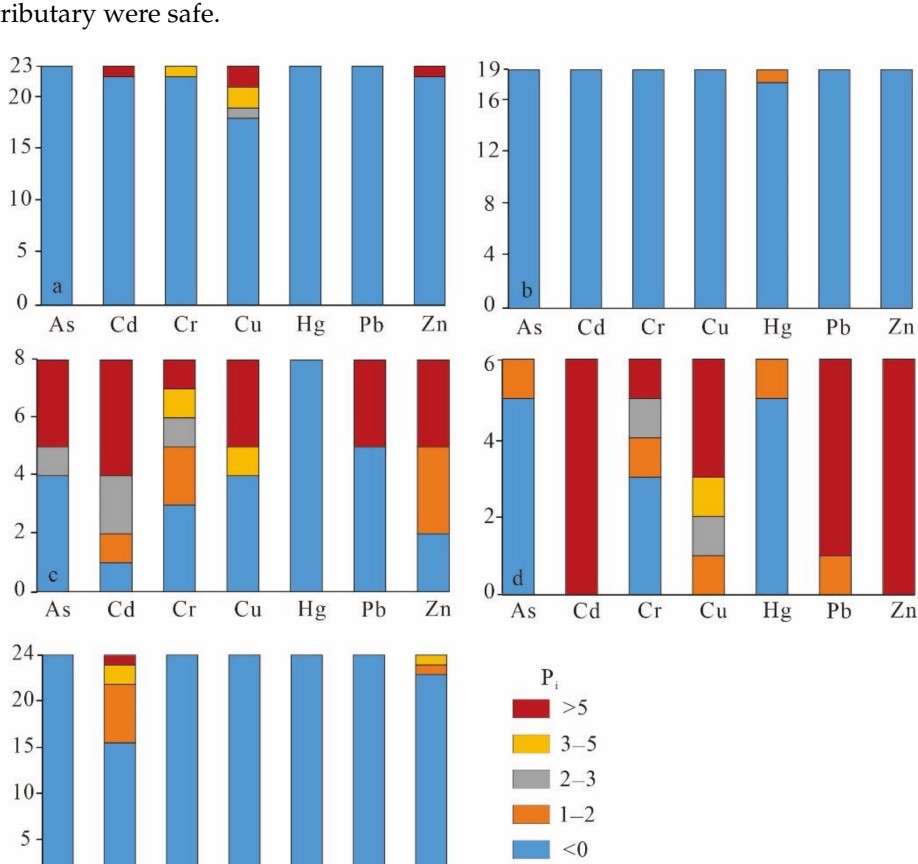

**Figure 7.** Number of $P_i$ levels for surface water samples in different streams: (**a**) Dawu tributary, (**b**) Le'an River, (**c**) Fujiawu tributary, (**d**) Yinshan tributary, and (**e**) Dexing River.

3.3.2. Sediments

Figure 8a shows the proportions of different $I_{geo}$ levels of each HMs in the sediment samples. Notably, approximately half of the sediment samples were moderately polluted (6.5%), moderately-to-heavily polluted (13.5%), heavily polluted (15.0%), heavily-to-extremely polluted (9.0%), or extremely polluted (5.0%) by Cu. The As pollution reached the levels of moderate, moderate to heavy, heavy, heavy to extreme, and extreme accounting for 14.5%, 5.5%, 5%, 3.5%, and 2.5% of all samples, respectively. The pollution levels of Hg and Cd in the sediment samples were 6.0% and 8.0% of moderate polluted, 6.0% and 4.0% of moderate to heavy pollution, 5.0% and 0.5% of heavy pollution, 0.5% and 1% of heavy to extreme pollution, and 0.5% and 0.5% of extreme pollution, respectively. The proportions of all samples that reached moderate to extreme pollution ($I_{geo} > 1$) were 8.0% for Pb, 5.0% for Zn, and 2.0% for Cr, respectively.

Figure 8b shows the potential ecological risks of the HMs in sediments of the study area. Mercury in sediments posed a moderate, high, serious, and severe potential risk in ~19.5%, 10.5%, 4.5%, and 10.5% of the study area, respectively. Cu, Cd, and As in sediments posed moderate to severe risk ($E_{ir} > 40$) in ~37.5%, 37.5%, and 23.5% of the study area, respectively. Moreover, Pb posed a high risk in ~1.0%; however, all the sediments samples were free from ecological risk by Cr and Zn. The $RI$ values of the HMs in the sediments ranged from 36.72 to 9237.61. As shown in Figure 8c, 19.5% of samples were classified into moderate risk, 16.5% of samples were classified into high risk, and 13.5% of samples were classified into serious risk. By combining the result of $I_{geo}$ and $RI$, we can deduce that the predominant pollution of HMs in sediments originated from Cu, As, Hg, and Cd. Therefore, using multiple evaluation methods, we can obtain more reliable results.

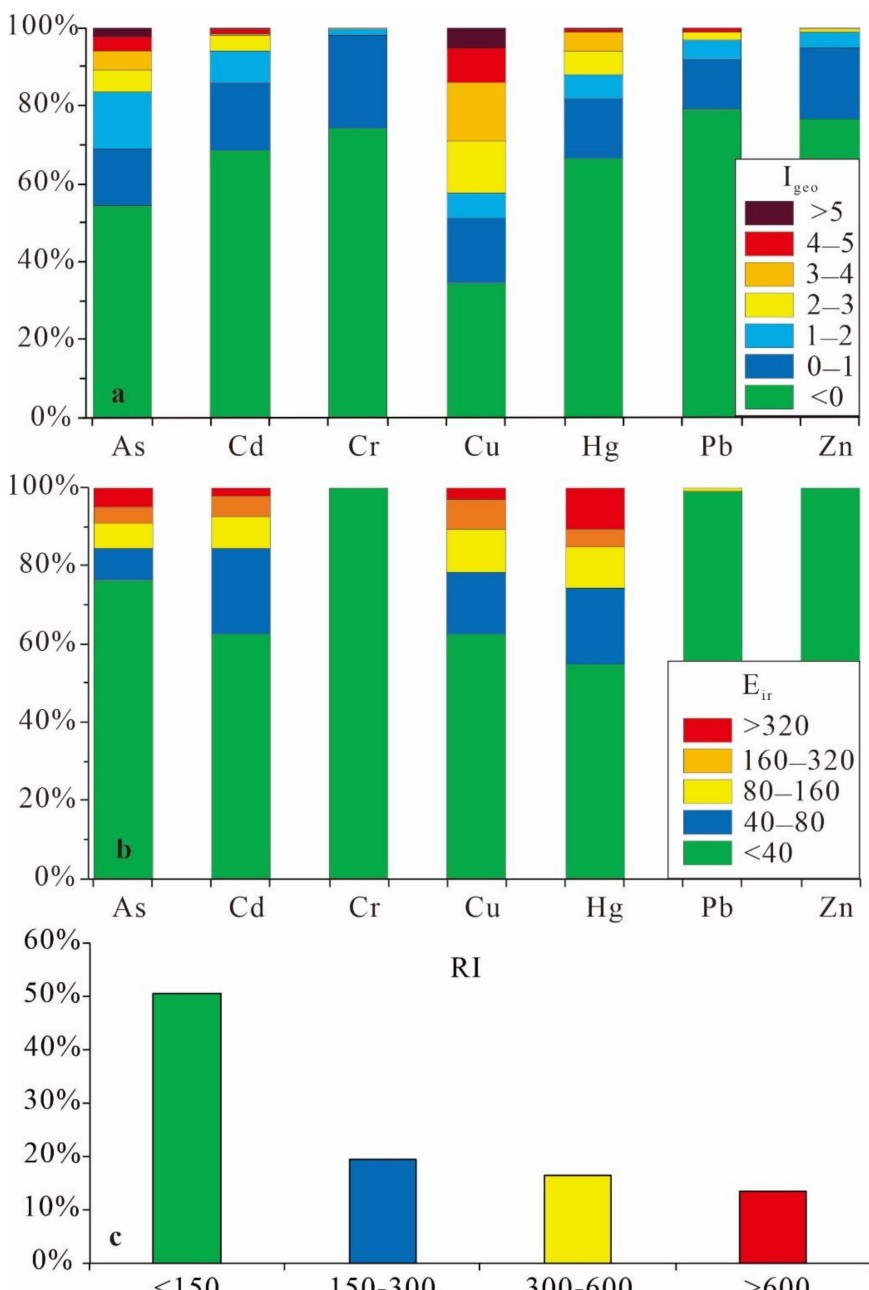

**Figure 8.** Proportions of different (**a**) $I_{geo}$, (**b**) $E_{ir}$, and (**c**) *RI* levels of HMs in all sediment samples (n = 200).

Figure 9 shows the potential ecological risks of the HMs in sediments of different streams. In particular, Cu, Hg, and Cd represented the major ecological risk elements in Le'an River and Dawu tributary, whereas Cu represented that in Fuxi tributary. The risk elements in Dexing River and Fujiawu tributary were As, Cd, Cu, and Hg, whereas those in Jinshan tributary were As, Hg, and Cd. The sediments in the mainstream exhibited similar ecological risk characteristics to its tributaries flowing through the mining area. Combined with the assessment results of surface water, the proportions of HM pollution in Dexing River and its tributaries were higher than those in Le'an River and its tributaries. Compared with other porphyry copper mine areas, the sources of HMs in sediments of Dexing area were more complex, and the ecological risks were higher [8,49]. The ecological risks of HMs in sediments were higher and wider, compared with the surface water in the study area.

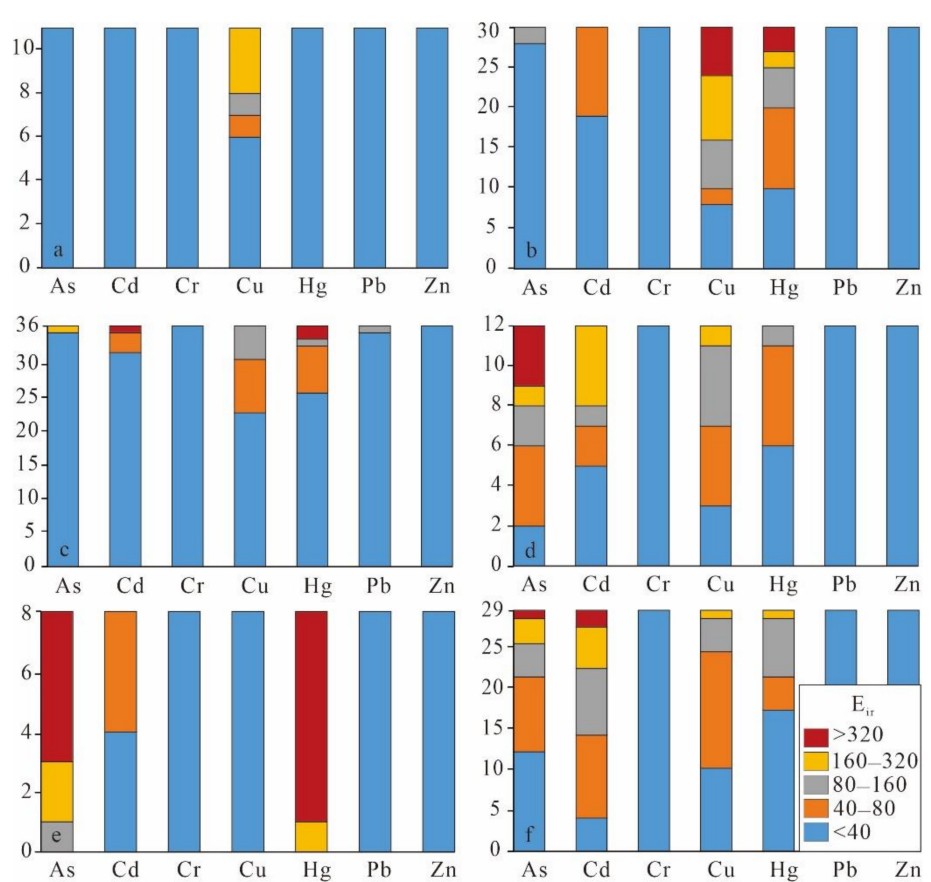

**Figure 9.** Number of $E_{ir}$ levels for sediments samples in different streams: (**a**) Fuxi tributary, (**b**) Dawu tributary, (**c**) Le'an River, (**d**) Fujiawu tributary, (**e**) Jinshan tributary, and (**f**) Dexing River.

## 4. Conclusions

The major objective of this study was to assess the levels of As, Cd, Cr, Cu, Hg, Pb, and Zn in ground water, surface water, and sediments. Thus, we elaborated the ecological risk of those HMs around the Dexing giant Cu-polymetallic ore cluster. The analysis indicates that the anthropogenic activity is the major driver of HMs pollutions in the study area. The degree of impact by anthropogenic activities followed the order sediment > surface water > ground water. Except the maximum value of As and Pb, the concentrations of the HMs in ground water were all lower than the threshold value of drinking water. The concentrations of HMs in the study area greatly varied, indicating that the surface water and sediments were severely disturbed by the mining activities. Meanwhile, the proportions of HMs pollution in the Dexing River and its tributaries were higher than those in Le'an River and its tributaries. The $P_i$, $P_n$, $I_{geo}$, and *RI* evaluations revealed that Cd, Cu, Zn, and Pb were the major pollutants in the surface water, and the major ecological risks of HMs in the sediments originated from Cu, As, Hg, and Cd. The surface water pollution levels were highest at tributaries closest to the mine area, whereas the sediment contamination extended several kilometers downstream of the major rivers. As a carrier of HMs in rivers, sediments are easily enriched in HMs which will be released to the external environment under specific conditions (e.g., acidification). Thus, the ecological risks of the HMs in sediments were higher and wider than that of the surface water in the study area. These results can provide a scientific basis for the tailored HM pollution management of water and sediments in the Dexing area. Future study should focus on the combination of numerical estimation of the HM pollution with management solutions, which can pave the way for future environmental remediation strategies in mines in the Dexing area.

**Author Contributions:** Conceptualization, methodology: H.P.; data curation, formal analysis, writing-original draft preparation: H.P. and R.Y.; supervision, project administration and writing-review and editing: G.Z. and Z.C.; investigation, supervision and data curation: B.S. All authors have read and agreed to the published version of the manuscript.

**Funding:** This research was funded by China Geological Survey, grant number 1212011087084. It was also supported by the Chinese Academy of Geological Sciences, grant number AS2019J03.

**Institutional Review Board Statement:** Not applicable.

**Informed Consent Statement:** Not applicable.

**Data Availability Statement:** Not applicable.

**Acknowledgments:** The authors are thanks to the anonymous referees for their perceptive comments and recommendations.

**Conflicts of Interest:** The authors declare no conflict of interest. The funders had no role in the design of the study; in the collection, analyses, or interpretation of data; in the writing of the manuscript, or in the decision to publish the results.

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
