# Peer review of "Heavy Metals and As in Ground Water, Surface Water, and Sediments of Dexing Giant Cu-Polymetallic Ore Cluster, East China"

_water, doi:10.3390/w14030352_

Round 1

Reviewer 1 Report

  1. The analytical protocol used for the study is sound. Results are well presented. Minor grammar check is required.
  2. Notation of units of measurement should be harmonized. Remove spaces between the forward slash and "L" (line 227-229). 
  3. The justification for the study in the introduction is weak. 
  4. The introduction should be revised with more recent papers (2019-2022). 

Author Response

Response to Reviewer 1 Comments

Point 1: The analytical protocol used for the study is sound. Results are well presented. Minor grammar check is required.

Response 1: Thanks for the comment. In the revised version, we have checked and modified the gramma of the full text, e.g. Line 12-13, 19, 22, 23-24, 28, 43, 56 and 125.

Point 2: Notation of units of measurement should be harmonized. Remove spaces between the forward slash and "L" (line 227-229).

Response 2: Thanks for the comment, the spaces between the forward slash and "L" have been removed in the revised version (Line 260-262).

Point 3: The justification for the study in the introduction is weak.

Response 3: We appreciate ‘your constructive suggestions. In the revised version, we have rewritten the justification for the study in Line 67-89.

Point 4: The introduction should be revised with more recent papers (2019-2022)..

Response 4: Thanks for the comment, the introduction has been revised more recent papers (Line 67-89 and 559-591).

Thank you again for your comments and suggestions.

Reviewer 2 Report

The work is very well done. The analysis of the research results is correctly presented, but the discussion of the results - comparisons with the results of other Authors - is missing. 
Small remark to writing units - they should be in SI system. 

Author Response

Response to Reviewer 2 Comments

Point 1: The work is very well done. The analysis of the research results is correctly presented, but the discussion of the results - comparisons with the results of other Authors - is missing.

Small remark to writing units - they should be in SI system.

Response 1: We appreciate your constructive suggestions, In the revised version, we have added some comparisons with the results of other Authors in “3.1. HMs Concentrations” (Line 240-242), “3.2. Spatial Distribution and Combined Characteristics of HMs” (Line 318-321, 325-326), and “3.3. Pollution Characteristics and Environmental Risk Assessment” (Line 471-473). Inaddition, we have checked and modified the units of the full text (Line 247, 262, 270, 272, 291, 297 and 301).

Thank you again for your advice.

Reviewer 3 Report

This study presents interest due to the obtained results regarding metal contamination of three different matrixes of ore cluster. The paper is well documented and structured, it presents the used methods and results in a logical and scientific way, all abbreviations are explained, as well as indicating its “utility” and also it seems to be that a lot of work and time were put into this article. Although, the novelty and originality were not mentioned.

Specific comments:

  1. In the title and manuscript, you can use only heavy metals and As or heavy metals and metalloids.
  2. Abstract should have no more than 200 words. Please check the Journal’s guideline as well.
  3. Line 26 – please correct kilometers.
  4. Please add the novelty of the study.
  5. Line 104 – please insert all the studied ponds, e.g. 1# - 3#.
  6. You could give more international flavor in the Results and Discussions section, in order to compare to other studies and understand the differences (HM results and pollution indices scores).

Author Response

Response to Reviewer 3 Comments

Point 1: In the title and manuscript, you can use only heavy metals and As or heavy metals and metalloids.

Response 1: Thanks for the comment, we have changed the “Heavy metal(loid)s” into “heavy metals and As” in the title and manuscript (Line 1, 11 and 35).

Point 2: Abstract should have no more than 200 words. Please check the Journal’s guideline as well.

Response 2: Thanks for the comment, we have revised the abstract within 200 words (Line 11-30).

Point 3: Line 26 – please correct kilometers.

Response 3: Thanks for the comment, we have corrected the spelling (Line 27).

Point 4: Please add the novelty of the study.

Response 4: Thanks for the comment, the novelty of the study have been added (Line 96-99).

Point 5: Line 104 – please insert all the studied ponds, e.g. 1# - 3#.

Response 5: Thanks for the comment, the 1# and 3# ponds (there is no 2# pond in the study area) have been insert in Line 134.

Point 4: You could give more international flavor in the Results and Discussions section, in order to compare to other studies and understand the differences (HM results and pollution indices scores).

Response 6: We appreciate your constructive suggestions, In the revised version, we have added some comparisons with the results of other Authors in “3.1. HMs Concentrations” (Line 240-242), “3.2. Spatial Distribution and Combined Characteristics of HMs” (Line 318-321, 325-326), and “3.3. Pollution Characteristics and Environmental Risk Assessment” (Line 471-473).

Thank you again for your advice.

Round 2

Reviewer 3 Report

Specific comments:

  1. Line 106 – You can insert Figure 1 after the first paragraph from page 3, in line 107.
  2. Line 160 – please correct AS with As.
  3. Line 257 – Try to fit the entire words and numbers in Table 2 e.g. “Loca-tion“, rang e” and the values for the pH.
  4. Lines 323, 325 and 327 – Please replace “Fig” with Figure, as in the rest of the titles of the figures.
  5. Line 337 – please replace “with” with to.

Author Response

Point 1: Line 106 – You can insert Figure 1 after the first paragraph from page 3, in line 107.

Response 1: Thanks for the comment. The Figure 1 has been inserted in Line 107.

Point 2: Line 160 – please correct AS with As.

Response 2: Thanks for the comment, the “AS” has been corrected.

Point 3: Line 257 – Try to fit the entire words and numbers in Table 2 e.g. “Loca-tion“, rang e” and the values for the pH.

Response 3: Thanks for the comment, The Table has been fited.

Point 4: Lines 323, 325 and 327 – Please replace “Fig” with Figure, as in the rest of the titles of the figures.

Response 4: Thanks for the comment, all “Fig” have been replaced with Figure.

Point 5: Line 337 – please replace “with” with to.

Response 5: Thanks for the comment, the manuscript havs been modified as suggested.

Thank you again for your comments and suggestions.